# Empirical Studies on Effect of Low-Level Laser Treatment on Glioblastoma Multiforme in Combination with Ag-PMMA-PAA Nanoparticles: Paired Red Region Optical-Property Treatment Platform

**Rohini Atluri [1], Daniel Korir [2], Tae-Youl Choi [1],* and Denise Perry Simmons [1,3],***

1   Department of Mechanical Engineering, University of North Texas, Denton, TX 76207, USA;
    rohiniatluri@my.unt.edu
2   Department of Chemistry, East Texas Baptist University, Marshall, TX 75670, USA; danielkorir@my.unt.edu
3   Department of Chemistry, University of North Texas, Denton, TX 76203, USA
*   Correspondence: tae-youl.choi@unt.edu (T.-Y.C.); denise.simmons@unthsc.edu (D.P.S.)

**Abstract:** Glioblastoma multiforme is an aggressive, invasive, fatal primary heterogenic brain tumor. New treatments have not significantly improved the dismal survival rate. Low-level laser therapy reports indicate different tumor cells respond distinctly to low-level laser therapy based on laser dose ($J/cm^2$) or with nanotherapeutics. We investigated the effects of pairing two optical property-driven treatment agents—a low-level laser on glioblastoma multiforme (U251) using an He-Ne laser (632.8 nm) with 18.8 nm spherical Ag-PMMA-PAA nanoparticles, with an absorbance peak at 400 nm with a broad shoulder to 700 nm. The He-Ne treatment parameters were power (14.87 ± 0.3 mW), beam diameter (0.68 cm), and exposure time 5 min leading to a 12.28 $J/cm^2$ dose. A dose of 12.28 $J/cm^2$ was applied to Ag-PMMA-PAA nanoparticle concentrations (110–225 μM). An amount of 110 μM Ag-PMMA-PAA nanoparticles combined with an He-Ne dose at 18 h yielded 23% U251 death compared to He-Ne alone which yielded 8% U251 death. A 225 μM Ag-PMMA-PAA nanoparticle He-Ne combination resulted in an earlier, more significant, U251 death of 38% at 6 h compared to 30% with 225 μM alone at 18 h. Both treatment agents possess inherent physical and functional properties capable of redesign to enhance the observed cell death effects. Our results provide evidence supporting next-step studies to test "the redesign hypothesis" that these paired optical-driven agents provide a tunable platform that can generate significant U251 cell death increase.

**Keywords:** low-level laser; He-Ne; glioblastoma multiforme; U251 cells; silver nanoparticles; brain tumor treatment

## 1. Introduction

### 1.1. Glioblastoma Multiforme

Glioblastoma multiforme (GBM) is one of the most lethal forms of malignant primary brain tumors killing approximately 18,000 Americans annually. According to the National Cancer Institute's (NCI) Surveillance, Epidemiology and End Results (SEER) program, new cases of brain and other nervous system disorders were 6.4% with a death rate of 4.4% per 100,000 persons within the same population in the same period. This statistic has remained steady since 1992 [1]. GBM accounts for 60–75% of all the astrocytic tumors and 12–15% of all brain tumors. Despite several decades of extensive research, surgical gains due to technological advancements, and aggressive combination treatments, the mortality rate attributed to GBM has not changed much. This calls for more effective long-term treatment options. If not treated, GBM is usually fatal within three months with an average survival time of one year following treatment after initial onset. Less than 34% of those diagnosed with GBM have a 5-year survival time even with the aggressive combined treatments of

surgery, radiation, and chemotherapy [1,2]. GBM is characterized by fast invasiveness into surrounding normal cells of the brain, chemo-resistance, radio-resistance, and acute vasculature which provides nutrients and oxygen to the cancer cells.

### 1.1.1. Carcinogenesis and Heterogeneity

Most brain tumors are known to be highly heterogeneous. Heterogeneity of GBM cells is attributed to clonal origins tracing back to molecular and cellular origins [2,3]. Each clonal population exhibits unique behavior different from that of other clonal subtypes. Furthermore, GBM cells can form stem cells (SC) with heterogeneous phenotypes that have unique responses to chemotherapeutic agents. This behavior poses a big challenge in developing an effective treatment regimen. One of the setbacks in glioma treatment posed by heterogeneity is the overlap in grades of carcinogenesis from normal to hyperplastic and dysplastic stages [4].

### 1.1.2. Existing Treatments

The current standard of care (SOC) for GBM treatment involves surgical resection followed by multimodal approaches, such as radiation (X-rays, gamma rays or photons) and adjuvant chemotherapeutics (for instance, temozolomide (TMZ) and bevacizumab). TMZ is an imidazotetrazine alkylating/methylating agent that binds to the N$-$ and O$-$ positions in the DNA strand leading to apoptosis in cancer cells [5]. However, cancer cells can quickly repair the damage caused by TMZ therefore limiting the efficacy of this drug and eventually leading to even more aggressive recurrence of a drug-resistant cancer. Furthermore, this treatment regimen comprising surgery, chemotherapy and radiation is often followed by severe neurological deterioration and poor quality of patient life after undergoing several aggressive treatment cycles [6,7]. There is, therefore, a need for more effective treatment that can lead to better treatment outcomes. With extensive research spanning several decades, many reports on new molecular targets and treatment responses have been reported (Table 1), but none has resulted in improved treatment outcomes [8].

**Table 1.** Select combination treatments in high grade GBM.

| Combination Treatments in High Grade GBM | Combination Therapy | FDA Approved | Treatment Indication | Proposed Mechanism of Action | Cross Blood Brain Barrier | Cross Blood Brain Tumor Barrier | Toxicity | Survival Benefit |
|---|---|---|---|---|---|---|---|---|
| Resection and TMZ | Yes | Yes | Both primary and recurrent | Induces cell cycle arrest at G2/M by DNA methylation at $N^7$ and $O^6$ on guanines and $O^3$ on adenines leading to apoptosis | Yes | No | Yes | Yes |
| Resection and Gliadel waferim- plants | Yes | Yes | Recurrent and primary high-grade glioma | Releases carmustine which modifies glutathione reductase to disrupt DNA function leading to apoptosis | Yes | Unknown | Yes | Yes |
| Resection, TMZ and Optune device | Yes | Yes | Recurrent and primary high-grade glioma | Delivers low-intensity, intermediate-frequency alternating electric fields that disrupts mitosis in tumor cells | Not applicable | Not applicable | Yes | Yes |
| Resection, LLL-He-Ne 632.8 nm and AgPMMA PAA | In vitro paired NIR window agents | No | In vitro primary origin brain grade 4 GBM | Apoptosis via caspase cascade by PARP cleavage at 63 kDa and 39 kDa | Yes | To be determined | In vitro human primary astrocyte cells -Yes | To be determined |

Recently, photothermal therapy (PTT), photodynamic therapy (PDT), and laser interstitial thermal therapies (LITT) have gained attention for treating brain tumors [9,10]. PTT is performed by selective local heating of a photothermal agent using a high-power laser with a maximum absorption rate in the near infra-red (NIR) region. PDT is performed by triggering site-specific photoactivated molecules called photosensitizers. This leads to a series of photochemical reactions, such as the formation of reactive oxygen intermediates responsible for damaging tumor cells [9].

LITT is performed by localized thermal ablation using a fiber-optic solid-state diode laser [10]. These therapies have limitations, such as the extent of thermal damage, thermal resistance of tumors, tumor recurrence and metastasis [9–11]. Thus, new translational research is needed to address the dismal performance of the existing treatments for patients diagnosed with GBM.

### 1.2. Low-Level Laser Therapy

In general, low-level laser therapy (LLLT) is known for its ability to enhance wound healing and tissue regeneration by manipulating the inflammatory, proliferative and remodeling phases of injury recovery [12]. Recent advances have suggested that low-level laser therapy (LLLT), also known as photo-biomodulation (PBM), is based on the dose measured in joules per square centimeter ($J/cm^2$). PBM has the potential to alter tumor behavior, such as proliferation or invasion. [13]. LLLT is performed by exposing the diseased cells to low levels of red and NIR light. The biochemical pathway associated with LLLT is unclear. Some reports suggest that the low-level light is absorbed by cytochrome c oxidase in mitochondria, which, in turn, alters ATP production, reactive oxygen species (ROS) and transcription factors, such as NFκβ. The optical window of LLLT falls from the red to the NIR region (600–1070 nm), and the power ranges between 1–1000 mW depending on the application [12,13]. Helium-neon (He-Ne) laser (λ = 632.8 nm) falls in the red region and can penetrate up to 1 mm enabling delivery of the highest percentage of incident energy to a specific volume of tissue [14,15]. In addition to its wound-healing properties, LLLT is being considered in cancer treatment when used at fluence parameters of power density ($mW/cm^2$), and irradiation time (seconds) [13,16,17]. LLLT showed no signs of transformation in normal cells while 808 nm laser irradiation with a dose of more than 5 $J/cm^2$ inhibited cell proliferation in GBM cells in vitro [13,18]. Over the last five years, studies have focused on elucidating the mechanism of combination treatments, such as nanoparticles (NPs) and chemotherapeutics, chemotherapy and LLLT, or a regimen of different chemotherapy drugs which may have different mechanisms of action in cell death [19,20].

The Role of Nanoparticles in Combination with LLLT

Interest in using metallic NPs in cancer treatment has continued to increase in the last decade [21]. A recent shift in the many uses of silver NPs as an anti-cancer agent has emerged [22,23]. Among the anticancer effects of silver NPs are activity in inducing cell death via generation of reactive oxygen species (ROS), weakening of colony-forming ability, inducing apoptosis and inhibiting the migratory behavior of cells [24–26]. AgNP has been found to be more cytotoxic to cancer cells than normal cells by making the cancer cells more susceptible to ROS-mediated apoptosis, including observed increases in sensitivity of glioma cells to chemotherapy drugs, such as TMZ [27].

The manipulation of ultra-low fluences at 17 $\mu W/cm^2$ and 1.5 $J/cm^2$ during photodynamic therapy has been shown to achieve nearly 100% cell killing of glioma cells over a two-week period [28]. Application of a photosensitizer, which is activated by light of a specific wavelength, has been used after surgery. This combination has been found to kill infiltrative cancer cells surrounding the tumor cavity and been shown to induce apoptosis of cancer cells by releasing ROS [16,28].

In this study, we designed an approach that paired two treatment agents with overlapping light-killing properties in the red region. We employed an extensive range of

red laser parameters and used varying concentrations of our specifically designed NIR-to red-range Ag-PMMA-PAA. This innovative combination was used to investigate its cell killing effect on U251 cells. The study combination showed promising results that support and warrant next-step "agent redesign" hypothesis studies to test for significantly enhanced U251 cell death.

## 2. Materials and Methods

### 2.1. Low-Level Laser Design and Characterization

A helium-neon laser (He-Ne; λ = 632.8 nm; Model 1135 P, JDSU) was used as a low-level laser in our experiments. The He-Ne laser beam diameter was expanded (Figure 1) to irradiate a monolayer of adherent cells in the entire well area of a 96-well plate in which experiments were conducted.

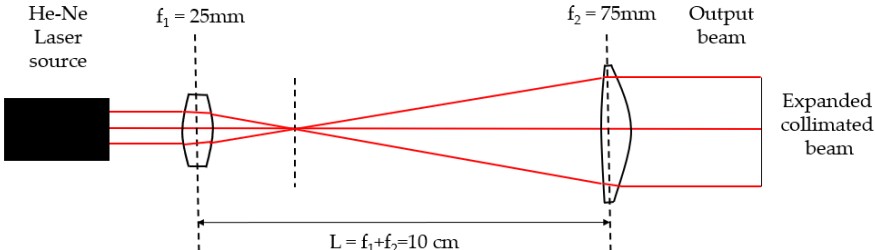

**Figure 1.** Helium-neon laser (λ = 632.8 nm) beam expansion setup.

The focal lenses, $f_1$ and $f_2$ at $L = f_1 + f_2$, were used with the beam magnifying $f_2/f_1$ times the input beam. The output beam diameter was measured by generating a Gaussian beam profile using the knife-edge method. The laser power of the expanded collimated beam was measured using a photodiode power sensor. The specifications were a wavelength range of 400–1100 nm, a power range of 50 nW–50 mW and an aperture size of 9.5 mm (Model S120 C, Thorlabs). The laser dose was determined using the following energy-density equation.

$$Dose \left( \frac{J}{cm^2} \right) = \frac{Power \ (W)}{Area \ (cm^2)} \times Time \ (s) \tag{1}$$

### 2.2. Synthesis of Ag-PMMA-PAA Nanoparticles

#### 2.2.1. Materials

Ultrapure water dispensed from a nanopure dispenser was used for all the reactions. A Spectrum 100 UVA lamp (300–400 nm) with 0–15 $W/cm^2$ power output with a fiber-optic light guide equipped with an IR filter and multicycle capability was used for photochemical synthesis of small nanoparticles. Borosilicate glass or quartz silica glass vials were used after thoroughly cleaning with deionized water followed by sterilization via autoclave prior to use. The product was collected in 1-cm-path-length quartz cuvettes and the absorption spectra were recorded using a Perkin-Elma Lambda double-beam UV/VIS/NIR absorption spectrophotometer to record the absorption spectra. Zeta potential and size measurements were analyzed using a Zetasizer Nano ZS series from Malvern Instruments. Centrifuge operations were performed using an Eppendorf 5810 R, a refrigerated centrifuge from Eppendorf AG, Germany. Microwave reactions were conducted in a Synthos 3000 from Anton Paar (Ashland, VA, USA).

#### 2.2.2. Synthesis of Ag-PMMA-PAA

The starting material of AgNO₃ (0.068 g, 20 mM, Sigma Aldrich, St. Louis, MO, USA) was dissolved in 20 mL of ultrapure water to make the first stock solution A. A PAA stock solution B was made by dissolving 0.74 mL (7.4 wt%) of medium molecular weight PAA (Sigma Aldrich) and made up to 10 mL using ultrapure water. A stock solution C was

prepared with fresh PMMA NPs which was prepared (procedure not included) using the established laboratory protocol [29]. A quantity of 1 mL of AgNO$_3$ solution was extracted using a micropipette and put in a fresh glass vial. Equal volumes of PAA and PMMA were added to the same fresh glass vial and the mixture was vortexed thoroughly for 5 min at ambient temperature. The mixture was then irradiated with a fiber-guided UV lamp for a total of 3 cycles with each cycle being 180 s, while gently stirring at ambient temperature to obtain a dark brown colloid of Ag-PMMA-PAA (Figure 2).

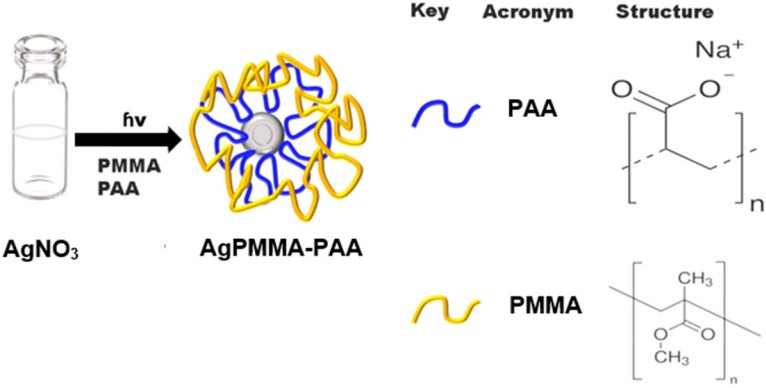

**Figure 2.** Scheme showing the synthetic protocol of the Ag-PMMA-PAA nanoparticle preparation, where silver nanoparticles were formed in PAA and PMMA.

Purification was carried out by dialysis at ambient temperature in the dark. An extract of 3 mL of the product was dialyzed using 3.3 mL/cm dialysis tubing with 6000–8000 molecular weight cut-off clamped at both ends and dialyzed for 40 h with one change at 24 h.

### 2.3. Cell Culture

U251 cells (Developmental Therapeutic Program (DTP) Repository, NCI Division of Cancer Treatment and Diagnosis) were cultured in Eagle's minimum essential medium without antibiotics, without phenol red and without L-glutamine (EMEM; Quality Biologicals) and supplemented with 10% fetal bovine serum (FBS; Gibco) and 2 mM L-glutamine (Sigma Aldrich, St. Louis, MO, USA). Human primary astrocyte (HPA) cells (Birth Defects Research Laboratory, University of Washington) were cultured in 1:1 *v/v* DMEM:F12 medium (Gibco) and supplemented with 10% fetal bovine serum, 1x penicillin-streptomycin-neomycin (PSN; Sigma Aldrich) and Fungizone (Sigma Aldrich). All cultures were maintained at 37 °C, 5% CO$_2$ and 95% relative humidity. The cells were washed with Dulbecco's phosphate-buffered saline (DPBS) without calcium and magnesium (Gibco) and were trypsinized with 0.25% trypsin-EDTA without phenol red (Sigma Aldrich) for sub-culturing and plating for experiments. Experiments were conducted with cells that were in the doubling phase and subcultures did not exceed passage seven. The subculture passage number began at P1, when the frozen repository stock vial was seeded onto a flask. It is known that the GBM tumor itself is inherently composed of heterogeneous clonal populations [2,30], so we had no control over the populations being frozen in each vial, except for the passage number, date of freezing, and reference to light microscopy images taken during subculture for freezing, as well as images taken during the experiments. Thus, each GBM culture might differ in composite cell populations, morphological phenotype, and treatment response across experiments.

### Cell Viability Assay

A CellTiter-Glo (CTG) luminescent cell viability assay was performed after the treatments to assess cell viability. At the end of various treatments, 25 µL of CellTiter-Glo 2D solution (Catalog #G9242, Promega, Madison, WI, USA) was added to each well and was incubated on a temperature-controlled orbital shaker for 2 min at 37 °C; the samples were

read by a microplate reader (Synergy 2, Biotek, Winooski, VT, USA) as the peak emission for CTG is 560 nm.

### 2.4. Statistical Analysis

All the data were expressed as the mean ± standard deviation (SD) of experiments in at least triplicates. All the data were analyzed using a one-way (cytotoxicity assays) ANOVA with Fisher's least significant difference test using SPSS Statistics 25.0.0. Differences were considered statistically significant with $p < 0.05$.

### 2.5. Experiments

#### 2.5.1. Experimental Set-Up

All the treatments were carried out in 96-well plates (Catalog #7342327, VWR) and in triplicate. The U251 cells were seeded at a concentration of $10^4$ cells/well in 100 μL media and were cultured for 24 hrs. The HPA cells were seeded at a concentration of $5 \times 10^4$ cells/well in 100 μL media and were cultured for 24 h. The 0 h time point started immediately after 24 h of culture, when the cells were in the doubling phase. The cells without any treatment were considered as a negative control. The cells which were killed by placing in a water bath at 70 °C for 60 min prior to the experiments were considered as a positive control. Silver nitrate ions of the same concentration as Ag-PMMA-PAA nanoparticles (AgPP NPs) were used as a control for assessing any silver ion toxicity effect. A protocol combination treatment order of AgPP NP followed by He-Ne was based on the AgPP NP absorption peak near 400 nm extending to 700 nm, and the He-Ne laser 632.8 nm. Post-treatment assessment for cell viability changes were performed immediately after He-Ne dose.

#### 2.5.2. He-Ne Laser Treatment Alone

At 0.5 h incubation, the U251 and HPA cells were irradiated with He-Ne laser at a dose of 12.29 J/cm² (using Equation (1)), which takes into account an exposure time of 5 min; treatment was immediately followed by cell viability analysis to determine the post-treatment % change in cell viability. Post-treatment cell viability assessment continued across a 24-h exposure time. The irradiation treatment was performed at 0.5 h, 6 h, 18 h, and 24 h, immediately followed by cell viability analysis at respective time points. The % decrease or % increase in cell viability was analyzed with respect to the negative control at respective time points. All the laser treatments were performed with a 96-well plate on an aluminum bead bath at 37 °C and in the dark, as shown in Figure 3. The distance between the output-collimated laser beam and the bottom of the 96-well plate was maintained at 10 cm to ensure constant depth of focus throughout the experiments.

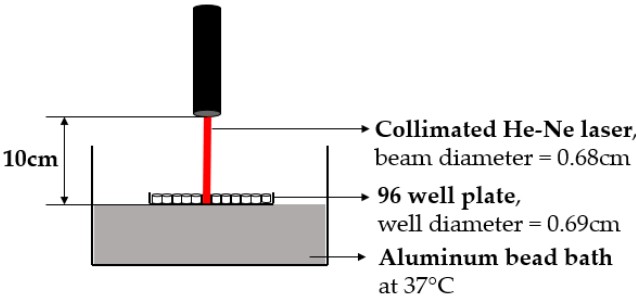

**Figure 3.** He-Ne Laser treatment experimental set-up.

#### 2.5.3. He-Ne Laser and AgPP NP Combination Treatment

The same synthesized and characterized AgPP NP original stock was used throughout the study. Serial dilution was set-up to achieve different concentrations of treatment stock AgPP NPs. The U251 cells and HPA cells were treated with increasing final concentrations (110 μM, 150 μM and 225 μM) of AgPP NPs, for 0.5 h, 6 h, 8 h and 24 h. Immediately after

these incubation hours, the cells were irradiated with an He-Ne laser dose of 12.29 J/cm$^2$, followed by assessment for cell viability; % decrease in cell viability was analyzed with respect to the negative control at respective time points. In addition, the U251 cells and HPA cells were treated with AgPP NPs alone with increasing final concentrations (110 μM, 150 μM and 225 μM) of AgPP NPs, for 0.5 h, 6 h, 18 h and 24 h, immediately followed by cell viability analysis. As a control for unreacted silver ions in the AgPP NPs, the cells were treated with silver nitrate ions (Ag ions) of the same concentration as AgPP NPs for the same exposure times and were assessed for cell viability. All experiments were performed in parallel.

## 3. Results

### 3.1. He-Ne Laser Characterization

The He-Ne laser beam was expanded to an output beam with a $1/e^2$ diameter of 0.68 cm, measured using the knife-edge method, as shown in Figure 4. The resulting exposure area was 0.363 sq. cm. The output beam remained collimated up to 15 cm from the 75 mm focal lens (Figure 1). The laser power was measured to be $14.87 \pm 0.3$ mW.

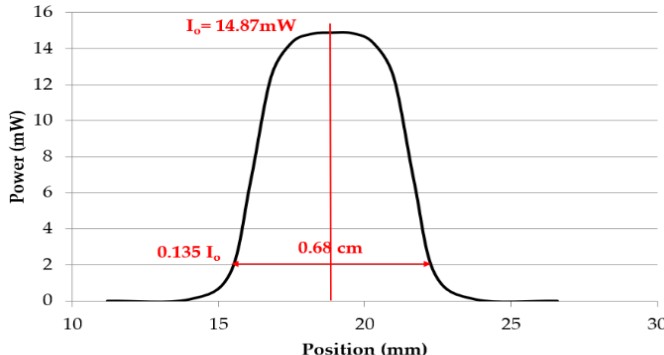

**Figure 4.** Gaussian curve of He-Ne laser (λ = 632.8 nm) using the knife-edge method.

### 3.2. Ag-PMMA-PAA Nanoparticle (AgPP NP) Characterization

Plasmonic nanoparticles, such as silver, possess unique optical properties determined by their size, shape, and surface plasmon resonance. These NPs induce absorbance maxima. The synthesized AgPP NPs were subjected to a set of physical characterization procedures. First, the formation of small anisotropic nanoparticles was shown by the presence of a plasmonic absorption peak near 400 nm. The absorbance shoulder stretched until the red region (around 600 nm). The absorbance was higher around 600 nm relative to the NIR region (around 780 nm) thus allowing the excitation of AgPP NPs when exposed to the 632.8 nm He-Ne laser (Figure 5a).

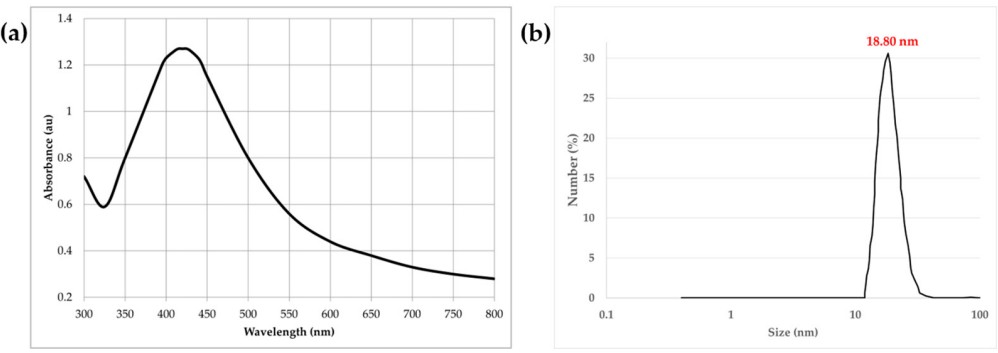

**Figure 5.** Physical characterization of AgPP NPs: (**a**) UV/Vis spectra for AgPP NPs with peak maxima at 400 nm with a broad shoulder stretching to near 700 nm (left) and (**b**) size distribution curve.

A Malvern Zetasizer Nano (MZN) was used to generate DLS-characterized AgPP NP size and shape (DLS as an indicator of particle shape.pdf; www.Malvern.com accessed on 7 February 2022). DLS revealed an AgPP NP average size of 18.8 nm (Figure 5b). The stability of the AgPP NPs was assessed by measurement of the zeta potential, at $-50.4$ mV. A nanoparticle zeta potential > $(+/-)$ 30 mV is considered stable and is characteristic of small particles capable of resistance to agglomeration, and thus is an indicator of stable AgPP NPs. Figure 5 shows the UV/Vis absorbance spectra for AgPP NPs nanoparticles. This indicates the formation of small spherical AgPP NPs for which the absorbance stretched to nearly 700 nm before baselining. The MZN was used to extract a non-visual output of the AgPP NP shape; this output was calculated using the DLS-measured hydrodynamic radius. The size of these NPs was confirmed via DLS, which showed an average size of 18.8 nm with a polydispersity index (PDI) of 0.388 and a surface charge of $-50.4$ mV (Table 2) that was attributed to the stabilizer PAA. The PDI value of AgPP NP was within the ISO standards for monodispersity (0.05–0.7), which was attributed to the spherical shape of the AgPP NPs [31]. The MZN DLS-based data agreed with the UV/Vis data. Both UV/Vis and MZN-DLS calculations supported the interpretation that the AgPP NPs were small spheroidal particles. The next steps would be image confirmation using SEM and/or TEM.

**Table 2.** Zeta potential measurements with size and their standard deviations.

|  | **Dialyzed (40 h)** | **Size (nm)** |
|---|---|---|
| Zeta potential (mV) | $-50.4$ | 18.80 |
| Std Dev | 7.37 | 7.93 |

*3.3. Low-Level-Laser-Treatment-Induced Changes in Cell Viability*

3.3.1. He-Ne Alone

When the U251 cells were irradiated with an He-Ne laser dose of 12.29 J/cm$^2$, 2–8%, a decrease in cell viability was observed. The time-dependent low-level laser (LLL) treatment exhibited a biphasic response, as shown in Figure 6.

The percentage changes in U251 cell viability observed at different post-treatment time points are summarized in Table 3.

**Table 3.** Cell death induced by low-level laser treatment in U251 cells.

| **U251 Cell Treatments** | **% Change in Cell Viability at Post-Treatment Analysis Time Points (Hours)** | | | |
|---|---|---|---|---|
| | **0.5** | **6** | **18** | **24** |
| Laser | 6 $^-$ | 1 $^{++}$ | 8 $^-$ | 2 $^-$ |
| 110 μM AgPP NP | 13 $^{++}$ | 5 $^-$ | 5 $^-$ | 26 $^{++}$ |
| 110 μM AgPP NP + Laser | 10 $^-$ | 15 $^-$ | 4 $^-$ | 15 $^{++}$ |
| 150 μM AgPP NP | 3 $^-$ | 5 $^-$ | 12 $^-$ | 20 $^{++}$ |
| 150 μM AgPP NP + Laser | 12 $^-$ | 3 $^-$ | 11 $^-$ | 12 $^{++}$ |
| 225 μM AgPP NP | 20 $^-$ | 28 $^-$ | 30 $^-$ | 5 $^-$ |
| 225 μM AgPP NP + Laser | 21 $^-$ | 38 $^-$ | 29 $^-$ | 11 $^-$ |

Note: In Table 3, ++ represents % increase in cell viability and – represents % decrease in cell viability.

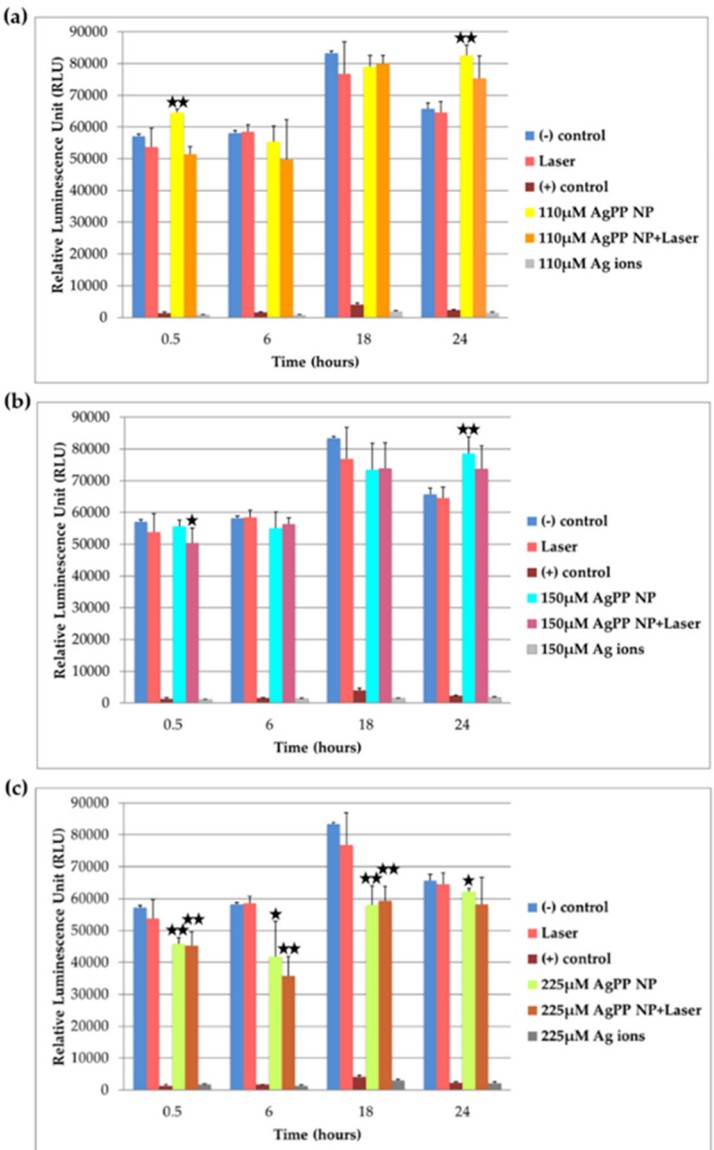

**Figure 6.** Time-dependent low-level laser treatment on U251 cells in combination with AgPP NPs at a concentration of (**a**) 110 μM, (**b**) 150 μM, and (**c**) 225 μM, respectively. ★ represents $p < 0.07$ and, ★★ represents $p < 0.05$.

### 3.3.2. AgPP NP Alone

During the 24-h post treatment window, using AgPP NP alone, the following U251 responses were observed. Increases in AgPP NP concentrations resulted in decreased U251 cell viability. The lowest concentration of 110 μM AgPP NP, resulted in U251 cell viability increases throughout the post-treatment window, from 13% at 0.5 h to a maximum of 26% at 24 h (Figure 6a). However, both the 150 μM and 225 μM AgPP NP individual treatments resulted in a biphasic response. A decrease in cell viability was observed up to 18 h with a significant U251 increase in viability at 24 h (Figure 6b,c). Of interest was the 225 μM AgPP NP significant response of a 30% U251 decrease in cell viability at 18-h post-treatment.

### 3.3.3. Combination AgPP NP and He-Ne Laser

At 110 μM AgPP NP followed by He-Ne laser dose, a biphasic response was again observed at 0.5 h and 24 h post-treatment relative to AgPP NP treatment alone. However, in sharp contrast to the increased U251 cell viability observed at 0.5 h and 24 h post-treatment with110 μM AgPP NP alone, the post-treatment of 110 μM AgPP NP, immediately followed

with laser irradiation at 0.5 h and 24 h, resulted in a significant decrease in cell viability of 23% and 11%, respectively. In addition, significant cell viability reduction was observed at 24 h.

At the 225 μM AgPP NP concentration, He-Ne laser addition resulted in an earlier U251 post-treatment response, which occurred at 6 h compared to the 18 h 225 μM AgPP NP alone post-treatment response. Moreover, the 6 h post-treatment response resulted in a 38% decrease in U251 cell viability compared to the negative untreated control at 6 h post-treatment and compared to the 18 h 30% U251 cell viability reduction post-treatment with AgPP NP alone.

The LLL treatment on HPA cells resulted in a 9% reduction in cell viability at a time of 24 h (Table 4).

**Table 4.** Cell death induced by low-level laser treatment in HPA cells.

| HPA Cell Treatments | % Change in Cell Viability at Post-Treatment Analysis Time Points (Hours) | | | |
|---|---|---|---|---|
| | **0.5** | **6** | **18** | **24** |
| Laser | 9 ++ | 5 − | 6 − | 9 − |
| 110 μM AgPP NP | 21 ++ | 15 − | 19 − | 13 − |
| 110 μM AgPP NP + Laser | 14 ++ | 11 − | 15 − | |
| 150 μM AgPP NP | 12 − | 35 − | 49 − | 15 − |
| 150 μM AgPP NP + Laser | 8 − | 31 − | 42 − | |
| 225 μM AgPP NP | 47 − | 54 − | 70 − | 76 − |
| 225 μM AgPP NP + Laser | 43 − | 61 − | 59 − | 91 − |

Note: In Table 4, ++ represents % increase in cell viability, − represents % decrease in cell viability and indicates data unavailable.

The concentration-dependent AgPP NP treatment resulted in decreased cell viability in HPA cells with a maximum 76% decrease in cell viability post-225 μM AgPP NP treatment at 24 h. However, the laser treatment, in combination with AgPP NPs, increased the cell viability of HPA cells while the AgPP NPs alone decreased cell viability.

Both HPA and U251 cells were minimally affected by the He-Ne laser irradiation dose alone (Figure 7a). HPA cells were found to be more sensitive to AgPP NP treatments when compared to U251 cells (Figure 7b). However, U251 cells proved to be sensitive to AgPP NP in combination with laser irradiation with an opposite effect on HPA cells (Figure 7c).

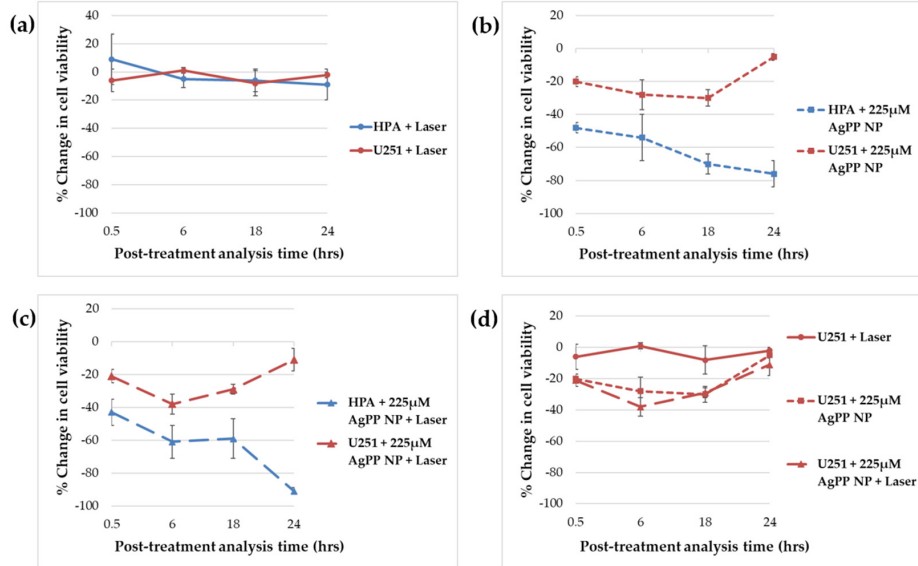

**Figure 7.** Comparison of percentage change in cell viability of HPA and U251 cells upon treatment with (**a**) laser alone, (**b**) 225 μM AgPP NP alone, (**c**) 225 μM AgPP NP in combination with laser. (**d**) cumulative GBM chart comparing treatments (**a–c**) for U251 cells.

## 4. Discussion and Conclusion

### 4.1. Research Advantages of the Selected Two Overlapping Red-Region-Based Treatment Agents

Pairing of these two specific optical property-driven treatment agents enables leveraging of prior researchers' and our current research evidence-based next-step studies. Figure 5a shows an absorption peak near 400 nm. This absorbance shoulder stretched until the red region (around 600 nm) and the absorbance was higher around 600 nm relative to the NIR region (around 780 nm) allowing the excitation of AgPP NPs when exposed to the 632.8 nm He-Ne laser. Therefore, functionalizing the AgPP NPs to target U251, followed by He-Ne, would be an agent redesign. For this type of redesign, we anticipate in vitro significant treatment-induced reduction in cell viability and cytotoxicity. Such results would lead to in vivo studies to demonstrate significantly improved survival benefit and decreased treatment-related toxicity. Moreover, these overlapping red-region paired optical-property-driven treatment agents each possess inherent separate manageable killing properties: the He-Ne laser and three variables controlling the dose in Equation (1), and the AgPP NP redesign agent delivery chemistry (functionalization) and the in-house Ag-PMMA synthesis addition of PAA, stabilizing, protecting, and increasing the available activity time for the Ag-PMMA.

The discussion highlights several promising observation-based next-step hypotheses and our proposed conclusions.

### 4.2. The GBM U251 Biphasic Response

The GBM U251 biphasic response observed could be due to heterogenic clonal populations of malignant glioma (U251) cells. Studies that use light microscopy can capture morphology during the treatment window and provide a first assessment of proposed cell population changes, and histology staining allows for a more definitive study of cell morphology. Bigner et al. have published extensive GBM images using light microscopy of histology [30]. Cell morphology studies can also drive and support treatment effects and cell response resistance and direct killing sensitivity hypotheses in heterogeneous cell populations and accompanying unexpected shifts in proliferation profiles. A treatment-induced cell-sensitizing-effect hypothesis could explain the window of opportunity for combination synergy, as observed in Figure 7d, whereby an He-Ne laser may allow a window of time for PAA-stabilized Ag-PMMA, being longer-acting for the desired U251 death effect using AgPP NPs. The increase in cell viability suggests that a population of cells have broken though the treatments at 18 h and, among several hypotheses, there is an emergence of a new population of cells resistant to the treatments. The inclusion of light microscopy images as an inherent part of the post-treatment in the experimental study design will prove informative for interpretations regarding unexpected results or population shift assessment in heterogenic tumors.

### 4.3. HPA Increase GBM Decrease in Cell Viability

Investigating AgPP NP redesign to deliver an agent that specifically targets GBM cells can address two related observations (Figure 7c): the HPA significant increase in viability concomitant with a GBM significant decrease in cell viability. Another consideration is the early assessment of our proof-of-concept AgPP NP IC50. IC50 values can help understand the efficacy of a treatment and initiate discussion on the control of excess AgPP NP and off-target activity. We calculated IC50 295 μM AgPP NP with He-Ne Laser and 336 μM AgPP NP alone. We used the 6 h post-treatment time point, which delivered the strongest treatment effect on the GBM cells. In this instance, the data suggest that, when used in combination with the He-Ne laser, a lower dose of AgPP NP can be used to kill 50% of the GBM cells. It is important to point out that we believe these IC50 concentrations are limited by the known heterogeneity of the GBM cell line at any given point in time. We have proposed investigation of heterogeneity to explain the observed dose-response during the 24-h post treatment window. We are also aware that the in vitro microenvironment does not reflect the in vivo microenvironment, although the GBM tumor mass is heterogeneic.

AgPP NP cargo functionality might require multiple-agent delivery design during the 24-h post-treatment.

*4.4. Increasing the Cell-Death Effect on GBM*

In addition to AgPP NP cargo delivery chemistry redesign, the He-Ne laser dose can be manipulated using Equation (1) and selection of a time increase would change the fluence, and, thus, in this example, increase the dose. It should be noted that our studies have demonstrated 30% and 38% increases in cell death within 18-h and 6-h post-treatment, respectively, while studies reporting 100% killing with Ag NPs were undertaken over a two-week period [28].

*4.5. Inherent Challenges to Expected Outcomes*

It is important to keep in mind that the heterogeneity of GBM does not allow a guarantee of homogeneous sample-to-sample cell populations which can affect the sample replicate outcomes, expected treatment response curves, and the statistical analysis. Morphology image inclusion as part of experimental study design can help provide some measure of explanation.

Nonetheless, our paired, optically driven platform using He-Ne, AgPP NP represents a potential tunable, modular treatment platform. Moreover, the inherent agent re-design possibilities for He-Ne laser and AgPP NP can influence improved treatment outcomes. In this regard, we have expertise in metallic polymeric NP size tunability [32], Ag-PMMA cargo delivery with PAA synthesis stability for longer circulating times [29], and the use of light to treat central nervous system heterogeneic brain disorders [33].

*4.6. The Metallic Nanoparticles Toxicity Debate: Gold versus Silver and Our AgPP NP*

Dr. Korir has studied both Au NP and Ag NP. We are interested in investigating more treatment options for patients. It was thought that Ag ions were the cause of toxicity. Dr. Korir has provided earlier evidence that his synthesis and purification protocol does not yield toxicity on treatment in cell culture. He further tested this concern with the AgPP NP in these GBM studies and showed that toxicity due to Ag ions was not a factor. With this Ag ion hurdle overcome, the Ag nanoparticles already discussed and the referenced reports on AgNP antibacterial properties are convincing. Moreover, it is our intention to contribute to the body of ongoing studies that address designer AgNP. Key to our advancing AgPP NP testing as a new proof-of-concept treatment in GBM are the results from such articles as the February 2022 report on enhanced AgNP hemocompatibility [34] and the June 2020 mouse model in vivo bladder cancer study [35]. In summary, we are encouraged by our proof-of-concept results, which support in vitro studies for a next-step testable "agent redesign hypothesis" (ARG). The ARG is expected to demonstrate that our tunable paired optical-property agents can generate significantly improved U251 cell death. This improvement is expected to occur without cytotoxic side-effects and lead to subsequent in vivo studies that demonstrate significantly increased survival rates and significantly reduced treatment toxicity. Should the "agent redesign hypothesis" deliver significantly improved U251 cell death, the paired optical-property approach can be extended to testing on other heterogeneic tumor masses, such as epithelial ovarian cancer.

**Author Contributions:** Conceptualization, T.-Y.C. and D.P.S.; methodology, D.P.S. and R.A.; validation, R.A., T.-Y.C. and D.P.S.; formal analysis, R.A.; investigation, R.A.; resources, T.-Y.C. and D.P.S.; data curation, R.A.; writing—original draft preparation, R.A.; writing—review and editing, D.P.S. and T.-Y.C.; visualization, D.K.; supervision, T.-Y.C. and D.P.S.; project administration, T.-Y.C.; funding acquisition, T.-Y.C. All authors have read and agreed to the published version of the manuscript.

**Funding:** This research was supported by a National Science Foundation, CBET award 1906553.

**Institutional Review Board Statement:** The study was conducted in accordance with the Declaration of Helsinki. The protocols fall under the following IRB approvals: (1) University of Washington Informed Consent 11/17/17ver32—Clinical Consent with Approval Stamped "11/20/2017 UW HSD IRB" for which a copy is on file with our journal submission, (2) University of North Texas IRB# 19-588, Approval 27 November 2019 and IRB #19-690, Approval 11 December 2019.

**Data Availability Statement:** Not applicable.

**Acknowledgments:** The human glioma (U251) cell line was received as part of a Cancer Models in Translational Research proposal. The MTA recipient is UNT investigator Denise Perry Simmons. and the NIH-NCI provider is Michael M. Gottesman. The cancer cells were cryopreserved at the Developmental Therapeutic Program (DTP) Repository, NCI Division of Cancer Treatment and Diagnosis, having originally been obtained from the Pathology Laboratory of Duke University from Darell D. Bigner. Human primary astrocyte (HPA) cells, a healthy cell control and their growth media were received from the Pharmacology and Neuroscience Laboratory of Kathleen Borgmann, University of North Texas Health Science Center. Normal human astrocyte donor samples were obtained with permission from her active research collaboration with the Birth Defects Research Laboratory, University of Washington. We acknowledge the Chemistry Laboratory of UNT Mohammad Omary. Where most of the experiments were performed, including synthesis and characterization of nanoparticles and cell culture work. The Texas Woman's University, Chemistry and Biochemistry Laboratory of Rob Petros, graciously gifted us materials, including CellTiter-Glo for cell viability studies. UNT's Vladimir Drachev allowed us to use his Nano-plasmonics and Nano-optics Physics Laboratory for DLS studies. UNT's Francis D'Souza allowed us to use his light microscope.

**Conflicts of Interest:** The authors declare no conflict of interest.

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
