# Peer review of "Empirical Studies on Effect of Low-Level Laser Treatment on Glioblastoma Multiforme in Combination with Ag-PMMA-PAA Nanoparticles: Paired Red Region Optical-Property Treatment Platform"

_2673-3501, doi:10.3390/applnano3020008_

Round 1

Reviewer 1 Report

This research work is focused to research on the preparation of Ag-PMMA-PAA nanoparticles and their application as a therapeutic agent along with Helium-Neon laser against Glioblastoma Multiforme (GBM). The work reported in this manuscript is interesting and well presented. However, it requires corrections and improvements before acceptance. The work requires revision. Some comments are:

  1. Authors need to mention the full name of all the technical terms in the abstract.
  2. Authors have mentioned AgPP NP as “small spherical NPS” but there is no data or experiment to confirm that statement. SEM or TEM can be performed
  3. Calculate the IC50 values for Cell culture studied for AgPP NP treatment both with and without laser treatment.
  4. The authors have used two cells, U251 cells and HPA cells. In the abstract, no data is provided for HPA cells studies.
  5. There are many grammatical and sentence errors in the article, and the language organization needs to be improved
  6. Please add statistical tools used for data analysis at the end of the material and method section.
  7. Inconsistently in the representation like AgPP NP or Ag-PMMA-PAA

Reviewer 2 Report

This manuscript reported Ag nanoparticles for low level laser therapy of tumor cells. The authors comprehensively evaluated the influence of different parameters on anticancer efficacy. However, some issues are need to be addressed to further improve the manuscript.

  1. Why the authors chose Ag nanoparticles as the photothermal agent? Ag nanoparticles have higher toxicity than Au nanoparticles, which may not suitable for in vivo studies.
  2. The TEM images of Ag nanoparticles should be given.

Reviewer 3 Report

The manuscript entitled “Empirical studies on effect of low-level laser treatment on glioblastoma multiforme in combination with Ag-PMMA-PAA nanoparticles: Paired red region optical-property treatment platform”, by Atluri et al have very well presented a proof-of-concept study evaluating effect of low-level laser treatment with and without silver nanoparticles on GBM cell culture. The manuscript is written well. Please find some comments below as suggestions to improve this manuscript for the readers.

Comments for authors:

  1. Please include SD/error bars in Figure 7.
  2. Since the percentage change in cell viability is short-lived (~24 hours), what are your thoughts on the treatment regimen?
  3. Although we understand this is a proof-of-concept study, please share some insight and include statements to indicate the applicability of both treatments: laser penetration/focusing inside the brain, % of nanoparticles of such small size in brain compared to other regions of the body assuming i.v. administration?
